# Möbius Transformation for Fast Inner Product Search on Graph

**Zhixin Zhou, Shulong Tan, Zhaozhuo Xu and Ping Li**
Cognitive Computing Lab
Baidu Research
10900 NE 8th St. Bellevue, WA 98004, USA
1195 Bordeaux Dr. Sunnyvale, CA 94089, USA
{zhixin0825,zhaozhuoxu}@gmail.com, {shulongtan,liping11}@baidu.com

## Abstract

We present a fast search on graph algorithm for Maximum Inner Product Search (MIPS). This optimization problem is challenging since traditional Approximate Nearest Neighbor (ANN) search methods may not perform efficiently in the non-metric similarity measure. Our proposed method is based on the property that Möbius transformation introduces an isomorphism between a subgraph of $\ell^2$-Delaunay graph and Delaunay graph for inner product. Under this observation, we propose a simple but novel graph indexing and searching algorithm to find the optimal solution with the largest inner product with the query. Experiments show our approach leads to significant improvements compared to existing methods.

## 1 Introduction

This paper focuses on a discrete optimization problem. Given a large dataset $S$ with high dimensional vectors and a query point $q$ in Euclidean space, we aim to search for $x \in S$ that maximizes the inner product $x^\top q$. Rigorously speaking, we will develop an efficient algorithm for computing

$$p = \arg\max_{x \in S} x^\top q. \tag{1}$$

This so-called *Maximum Inner Product Search (MIPS)* problem has wide applicability in machine learning models, such as recommender system [35, 16], natural language processing [5, 33] and multi-class or multi-label classifier [38, 34], computational advertising for search engines [9], etc. Because of its importance and popularity, there has been substantial research on effective and efficient MIPS algorithms. The early work of [27] proposed tree-based methods to solve the MIPS problem. Recently, there is a line of works in the literature tried to transform MIPS to traditional *Approximate Nearest Neighbor (ANN) search* [11, 12, 18] by lifting the base data vectors and query vectors asymmetrically to higher dimensional space [2, 28, 30, 26, 29, 36]. After the transformation, the well-developed ANN search methods can then be applied to solve the MIPS problem. There are other proposals designed for the MIPS task including quantization based methods [15] and graph based methods [25].

In this paper, we will introduce a new graph based MIPS algorithm. Graph based methods have been well developed for ANN search in metric space and show significant superiority [20, 4, 24, 13]. The recent work [25], namely *ip-NSW*, attempts to extend the graph based methods for ANN search to MIPS. The authors introduce the concepts of *IP-Delaunay graph*, which is the smallest graph that can guarantee the return of exact solutions for MIPS by greedy search. Practically, ip-NSW tries to approximate the IP-Delaunay graph via *Navigable Small World (NSW)* [23] and *Hierarchical Navigable Small World (HNSW)* [24]. To improve upon existing methods, we propose a better graph based method for MIPS, which preserves the advantages of similarity graph in metric space.

Our method is based on *Möbius transformation* on the dataset, that connects graph based indices for MIPS and ANN search. We find that under Möbius transformation, there is an isomorphism between two graphs: (a) IP-Delaunay graph before the transformation. (b) A subgraph of the Delaunay triangulation w.r.t. $\ell^2$-norm ($\ell^2$-Delaunay graph) after the transformation. Based on this observation,

we approximate IP-Delaunay graph in two steps: (i) map the data points via Möbius transformation; (ii) approximate $\ell^2$-Delaunay graph on the transformed data points and one additional point for the origin. Afterward, given a query point, we perform a greedy search on the obtained graph by comparing inner product of the query with data points (nodes in the graph) in the original format.

The superiority of our method is two-fold: (a) The $\ell^2$-distance based graph construction can preserve all advantageous features of similarity graph in metric space; (b) The additional point (i.e., the origin) will be connected to diverse high norm points (usually solutions for MIPS), which will naturally provide good starting points for the greedy search. The empirical experiments demonstrate that these features significantly improve the efficiency.

## 2 Graph Based Search Methods and Our Approach

A graph based search method typically first constructs a well-designed similarity graph, e.g., $k$NN graph in Approximate Nearest Neighbor (ANN) search, then performs greedy search on the graph. *Simple greedy search*, such as for Maximum Inner Product Search (MIPS) task, can be described as follows. Given a graph and a query, the algorithm randomly selects a vertex from the graph, then evaluates the inner product of the query with the randomly seeded vertex and the vertex's neighbors. If one of its neighbors has a larger inner product with the query than the vertex itself, then we consider the neighbor as a newly seeded vertex and repeat the searching step. This procedure stops when it finds a vertex that has a larger inner product with the query than all the vertex's neighbors. Greedy search has a generalized version, which will be introduced in Algorithm 1 with more details.

It was pointed out in [1, 23] that in order to discover the exact solution of nearest neighbor search or MIPS by the greedy search strategy, the graph must contain the *Delaunay graph* (see Definition 2) with respect to (w.r.t.) the searching measure as a subgraph. For common ANN search cases, searching w.r.t. $\ell^2$-distance, the index graph should contain the Delaunay graph w.r.t $\ell^2$-distance (referred as $\ell^2$-*Delaunay graph*) as a subgraph. In practice, approximate $\ell^2$-Delaunay graphs are usually constructed due to the difficulty in building the exact Delaunay graphs, such as VoroNet [4] and Navigable Small World (NSW) [23]. Based on NSW, Hierarchical-NSW (HNSW) network [24] exploits the hierarchical graph structure and heuristic edge selection criterion (see Algorithm 3 for details), and often obtains performance improvement in ANN search tasks.

The idea of the Delaunay graph can be extended to inner product. The best graph for exact MIPS by simple greedy search is the Delaunay graph w.r.t. inner product (referred as *IP-Delaunay graph*). The recent work [25], namely ip-NSW, attempts to extend HNSW for metric spaces to MIPS. It is worth noting that the authors of [25] show some important properties of Delaunay graph. However, their HNSW based graph construction algorithm for inner product has some disadvantages:

1. Since the edge selection criterion of HNSW does not apply on inner product, the incident edges of a vertex can have very similar directions, which will reduce the efficiency.
2. The hierarchical graph structure of HNSW is helpful in ANN search for metric measures, but it has little effect on the MIPS problem.

We validate these claims by experiments on comparison with different versions of ip-NSW. The effect of edge selection can be positive or negative in different datasets. Hierarchical structure does not change the efficiency of inner product search. To resolve the edge selection issue, previously we proposed a proper edge selection method, IPDG, specifically for inner product [31]. IPDG improves the top-1 MIPS significantly but shows performance limitations for top-$n$ ( $n > 1$) results. In this

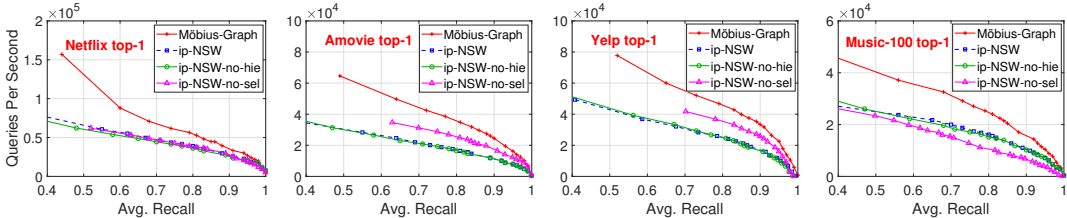

Figure 1: Experimental results for (top-1) Recall vs. Queries Per Second on different datasets. The curve on the top shows superiority of the corresponding method. Möbius Graph, ip-NSW, ip-NSW-no-hie, ip-NSW-no-sel stand for our proposed method, ip-NSW with both hierarchical structure and edge selection, ip-NSW without hierarchical structure, and ip-NSW without edge selection respectively.

paper, we propose a better approximation of IP-Delaunay graph (referred to as **Möbius-Graph**) for MIPS, which provides a state-of-the-art MIPS method for various top-$n$ MIPS results.

The intuition behind is that if we find a transformation that maps IP-Delaunay graph in the original space to $\ell^2$-Delaunay graph's certain subgraph in the transformed space, we can make full use of the successful $\ell^2$-Delaunay graph approximation methods to build an IP-Delaunay graph. Given each data point $x_i$, we perform the Möbius transformation $y_i := x_i/\|x_i\|^2$, from which we have a new data collection: $\tilde{S} = \{0, y_1, y_2, \ldots, y_n\}$. After Möbius transformation, we apply existing graph construction method (e.g., HNSW or "SONG" a recent variant [39]) and obtain an approximate $\ell^2$-Delaunay graph on the transformed data (i.e., $\tilde{S}$). We found that the IP-Delaunay graph w.r.t. $S$ is isomorphic to the neighborhood of 0 in $\ell^2$-Delaunay graph w.r.t. $\tilde{S}$. Details about this statement can be found in Section 3. In short, our approach can be summarized as the following steps:

1. Let $\tilde{S} := \{y_i = x_i/\|x_i\|^2 \mid x_i \in S\} \cup \{0\}$ be the transformed dataset.
2. Constructing approximate $\ell^2$-Delaunay graph, (e.g., HNSW), w.r.t. $\tilde{S}$.
3. Let $N$ denote the neighbors of 0 on the graph from the previous step. Then remove 0 and its incident edges from the graph, and replace the vertices $y_i$ by original data vectors $x_i$.
4. Let $N$ be initial vertices, then perform greedy inner product search on the graph.

Note that our greedy search algorithm starts from a set of initial points instead of the data point 0 since 0 is not in $S$. Multiple initial points are possible in generalized greedy search described in Algorithm 1. An equivalent description is starting from 0 but never returning it. Compared with the existing graph based search method for MIPS (i.e., ip-NSW), our approach builds the index graph by $\ell^2$-distance (on the transformed data), which can largely preserve advantageous features of metric similarity graph. Besides, our approach starts searching from well-chosen diverse top-norm points $N$ (the usage is similar as the hierarchical structure of HNSW), which will lead to more efficient performance. Therefore, our approach to a large extent overcomes the weakness of the existing graph based search method, and it is not surprising that our method performs empirically better.

## 3  Möbius Transformation and Delaunay Graph Isomorphism

As pointed out in [1] that, in order to find the exact nearest neighbor by simple greedy search, the graph must contain *Delaunay graph* as a subgraph. This statement can extend to MIPS problem [25]. For generality, we will introduce *Voronoi cell* and *Delaunay graph* for arbitrary continuous binary function $f : X \times X \to \mathbb{R}$; however, we are typically interested in the cases of inner product $f(x, y) = x^\top y$ and negative $\ell^2$-norm $f(x, y) = -\|x - y\|$ in this paper.

**Definition 1.** *For fixed $x_i \in S \subset X$ and a given function $f$, the Voronoi cell $R_i$ is defined as*
$$R_i := R_i(f, S) := \{q \in X \mid \forall x \in S, f(x_i, q) \geq f(x, q)\}.$$

Voronoi cells determine the solution of MIPS problem. One can observe from the definition above that, when $f(x, y) = x^\top y$, $x_j \in \arg\max_{x_i \in S} x_i^\top q$ if and only if $q \in R_j$. Since recording Voronoi cells is expensive. We instead record its dual diagram, namely *Delaunay graph*, defined as follows.

**Definition 2.** *For fixed function $f$ and dataset $S \subset X$, and given Voronoi cells $R_i$, $i = 1, 2, \ldots, n$ w.r.t. $f$ and $S$, the Delaunay graph is an undirected graph with vertices $S$, and the edge $\{x_i, x_j\}$ exists if and only if $R_i \cap R_j \neq \emptyset$.*

Delaunay graph records adjacency of Voronoi cells. If cell $R_i$ and cell $R_j$ is adjacent to each other, then there exists an edge between their corresponding nodes $x_i$ and $x_j$. If $f(x, y) = -\|x - y\|$, then the graph is called $\ell^2$-Delaunay graph. If $f(x, y) = x^\top y$, then the graph is called IP-Delaunay graph.

We now narrow the scope to MIPS problem. Let $f(x, y) = x^\top y$ and $X = \mathbb{R}^d \backslash \{0\}$, and we aim to solve the optimization problem (1). We remove 0 from $\mathbb{R}^d$ for two reasons. Firstly, 0 has the same inner product value with any points. Secondly, if 0 is not removed, then every Voronoi cell w.r.t. the inner product contains 0 as a common element, so the Delaunay graph will be fully connected and not interesting. We also require the following mild assumption on dataset to simplify the analysis.

**Assumption 1.** *The dataset $S$ satisfies that its conical hull is the whole space. More precisely,*
$$coni(S) := \Big\{ \sum_{i=1}^{n} \alpha_i x_i \Big| x_i \in S, \alpha_i \geq 0 \Big\} = \mathbb{R}^d. \tag{A1}$$

**Assumption 2** (General position). *For $k = 2, 3, \ldots, d + 1$, there do not exist $k$ points of the dataset $S$ lies on a $(k-2)$-dimensional affine hyperplane, or $k + 1$ points of $S$ on any $(k-2)$-dimensional sphere. If so, then we say dataset $S$ is in general position.*

Assumptions 1 and 2 are often mild in real data. When the data points are embedded vectors of users, items, (in recommender system) entities or sentence (in natural language processing). In these scenarios, the entries of data vectors are distributed on the whole real line. With high probability, each hyperoctant contains at least one data point so that the convex hull of the dataset contains 0 as an interior point. Assumption 2 holds with probability one if the data vectors in $S$ are independently and identically following any continuous distribution on $\mathbb{R}^d$. For such dataset $S$, the corresponding $\ell^2$-Delaunay graph and IP-Delaunay graph are unique. See [10] for details. Now we are ready to introduce two important criterion of these Delaunay graphs.

**Proposition 1** (Empty half-space criterion). *For a fixed dataset $S \subset \mathbb{R}^d$, suppose there exists an open half-space $H$ of $\mathbb{R}^d$ satisfying: (a) $x_i$ and $x_j$ are on the boundary of $H$, (b) $H$ contains no data points, then there exists an edge connecting $x_i$ and $x_j$ in IP-Delaunay graph. Conversely, if such an edge exists, then the open half space $H$ must exist.*

In other words, *empty half-space criterion* says, in IP-Delaunay graph, edge $\{x_i, x_j\}$ exists if and only if there is a $(d-1)$-dimensional hyperplane, which passes through $x_i$ and $x_j$, such that one of its corresponding open half-space is empty, and the other one contains all data points except $x_i$ and $x_j$. The empty half-space criterion of IP-Delaunay graph is closely related to *empty sphere criterion* of $\ell^2$-Delaunay graph as what follow.

**Proposition 2** (Empty sphere criterion). *For a fixed dataset $S \subset \mathbb{R}^d$, a subset of $d + 1$ points of $S$ are fully connected in the $\ell^2$-Delaunay graph corresponding to $S$ if and only if the circumsphere of these points does not contain any other points from the dataset $S$ inside the sphere.*

Once this criterion is satisfied, we call the subgraph of these $d + 1$ vertices a $d$-simplex. The proof of the empty sphere criterion is not provided here. We refer readers to see [14] for details. The connection between these criterions can be demonstrated by the transformation

$$g : \mathbb{R}^d \backslash \{0\} \to \mathbb{R}^d \backslash \{0\}, \quad g(x) = \frac{x}{\|x\|^2}. \tag{2}$$

Under this transformation, every hyperplane is mapped to a sphere passing through the origin. This is due to the fact that transforms on $\mathbb{R}^d$ of the form

$$g(x) = b + \frac{A(x - a)}{\|x - a\|^\epsilon} \tag{3}$$

for orthogonal matrix $A$ and $\epsilon = 0$ or 2 are *Möbius transformations*. Indeed, by Liouville's conformal mapping theorem (a generalized version can be found in [21]), for $d > 2$, (3) characterizes all Möbius transformations. An important and useful property of Möbius transformation says, if a hyperplane does not pass through origin, then its image under any Möbius transformation is a sphere passing through the origin.

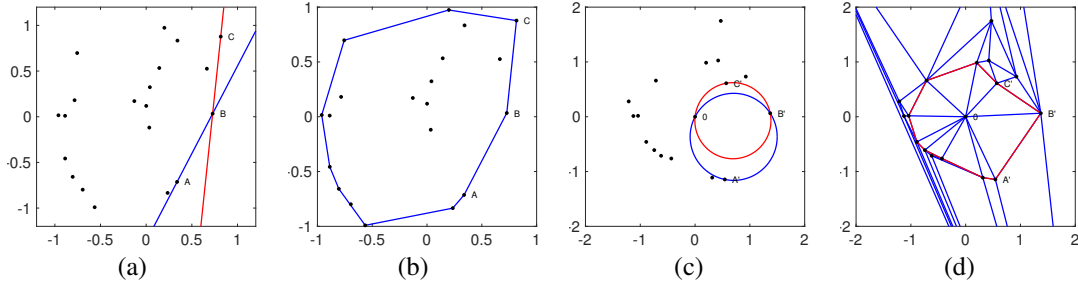

Figure 2: (a) Empty half-space criterion for IP-Delaunay graph. (b) The IP-Delaunay graph. (c) Empty sphere criterion for $\ell^2$-Delaunay graph after transformation. (d) The $\ell^2$-Delaunay graph after transformation. The red edges form the subgraph that is isomorphic to IP-Delaunay graph.

Figure 2 shows an example when $d = 2$. The line $AB$ in Figure 3 divides the plane into two open half-spaces. One of the half-space does not contain any data points, so $A$ and $B$ are connected in IP-Delaunay graph by Proposition 1. Let $A'$ and $B'$ be the images of $A$ and $B$ under transformation

(2). According to the property of Möbius transformation, the image of line $AB$ is the circumcircle of points $0$, $A'$ and $B'$ in Figure 3. The empty half-space criterion of $A$ and $B$ implies that the circumcircle does not contain any data points inside, so there is a simplex with vertices $0$, $A'$ and $B'$ in the $\ell^2$-Delaunay graph by empty sphere criterion. This observation is formalized as follows.

**Theorem 1.** *Let $X = \mathbb{R}^d \backslash \{0\}$. We assume $S$ satisfies Assumption 1 and 2. For $i \in [n]$, let $y_i := x_i / \|x_i\|^2$, $S' := \{y_1, \ldots, y_n\}$ and $\tilde{S} = S' \cup \{0\}$, then the following are equivalent:*

- *(a) The IP-Delaunay graph w.r.t. $S$ contains an edge $\{x_i, x_j\}$.*
- *(b) There exists $a \in \mathbb{R}^d \backslash \{0\}$ such that $x_i^\top a = x_j^\top a \geq \max\limits_{x \in S} x^\top a > 0$.*
- *(c) There exists $c \in X$ such that $\|y_i - c\| = \|y_j - c\| = \|c\| \leq \min\limits_{y \in S'} \|y - c\|$.*
- *(d) There exists a $d$-simplex in $\ell^2$-Delaunay graph w.r.t $\tilde{S}$ contains vertices $\{0, y_i, y_j\}$.*

Equivalence between (a) and (d) in the theorem implies an isomorphism between IP-Delaunay graph and a subgraph of $\ell^2$-Delaunay graph. Hence we immediately have the next corollary.

**Corollary 1.** *The following graphs are isomorphic after removing their isolated vertices:*

- *(a) the IP-Delaunay graph on $S$,*
- *(b) a subgraph of $\ell^2$-Delaunay graph on $\tilde{S}$ with every edge $\{y_i, y_j\}$ satisfying the following condition: there exists a $d$-simplex in $\ell^2$-Delaunay graph contains vertices $\{0, y_i, y_j\}$,*

*where the isomorphism is $x_i \mapsto y_i$ for $x_i$ that are not isolated in IP-Delaunay graph.*

Considering the example in Figure 2, Corollary 1 says the IP-Delaunay graph in Figure 3 is isomorphic to the subgraph in red in Figure 3. Thus, good approximation of $\ell^2$-Delaunay graph also applies to approximation of IP-Delaunay graph. See next section for implementation details.

**Remark 1** (Convex hull and extreme point). *If a vertex is not isolated in IP-Delaunay graph, then we say it is an extreme point. The concept of the extreme point is introduced in [3]. Under Assumption 1, a point is extreme if and only if it locates on the boundary of the convex hull of $S$. In this case, building the IP-Delaunay graph is equivalent to find the convex hull. In Corollary 1, we derive an equivalent way to find the convex hull of a finite set. For the purpose of convex hull construction, Assumption 1 is not required since it always holds after some translation. We note that there exist algorithms for finding convex hull [3]. This method is not computationally feasible on high dimensional data, and there does not exist a convex hull approximation in previous work, so we propose IP-Delaunay graph approximation by graph isomorphism in this paper.*

## 4   Implementation in Large High Dimensional Data

For large high dimensional data, finding the exact IP-Delaunay graph of the data points is not computationally feasible. Therefore, practical and efficient graph construction and searching algorithm for large scale data in high dimension are in demand. In this work, we provide the algorithm (summarized in Algorithm 4) for building **Möbius-Graph** and greedy search on it when we have massive high dimensional data. We will first introduce a generalized greedy search algorithm because it will be repeatedly used during graph construction and inner product search.

We recall that our goal of greedy search is to find $x \in S$ to maximize $f(x, q)$ for any query $q$. Here, we consider either $f(x, y) = -\|x - y\|$ or $f(x, y) = x^\top y$. For simplicity, we say the nearest neighbor of $x$ is $y$ when $y$ has largest evaluation of $f(x, \cdot)$. We first initialize priority queue $C$ (it can be random or well-chosen data points), then check the evaluation of $f(x, q)$ for all $x \in C$ and all out-neighbors of these $x$'s. Among those vectors we have evaluated, we replace $C$ by top-$k$ vectors in descending order of evaluation of function $f(\cdot, q)$. We consider the top-$k$ elements in $C$ as the new priority queue. We update $C$ until it does not change anymore. Algorithm 1 summarizes this procedure. If $k = 1$, then this generalized greedy search is equivalent to the simple version described in Section 2. This generalized greedy search allows the algorithm to return approximate top-$k$ items, which are valuable for query search and recommender system.

Now we are ready to present the graph construction algorithm (summarized in Algorithms 2). By Theorem 1 and Corollary 1, the best graph we want to use is IP-Delaunay graph on $S$, which is

---

**Algorithm 1:** GREEDY-SEARCH$(q, P, G, k, f)$

---

1: **Input:** query element $q$, a set of enter points $P$, graph $G = (S, E)$, number of candidates to return $k$, measurement function $f$.
2: Initialize the set of priority queue, $C \leftarrow P$.
3: Mark elements of $P$ as checked and the rest of vertices as unchecked.
4: **if** $|C| > k$ **then**
5:     $C \leftarrow$ top-$k$ elements of $x \in C$ in descending order of $f(x, q)$.
6: **while** $\exists x \in S$ unchecked and $C$ keeps update **do**
7:     $C \leftarrow C \cup \{y \in S : x \in C, y \text{ unchecked}, (x, y) \in E\}$
8:     Mark elements in $C$ as checked.
9:     **if** $|C| > k$ **then**
10:         $C \leftarrow$ top-$k$ candidates of $x \in C$ in descending order of $f(x, q)$.
11: **Output:** $C$.

---

---

**Algorithm 2:** GRAPH-CONSTRUCTION$(S, k, d)$

---

1: **Input:** dataset $S$, the size of priority queue $k$, maximum outgoing degree of graph $d$.
2: $n \leftarrow |S|$. For $i \in [n]$, let $y_i = x_i / \|x_i\|^2$.
3: $\tilde{S} \leftarrow \{0, y_1, \ldots, y_n\}$. Define $y_0 = 0 \in \tilde{S}$.
4: $G \leftarrow$ fully connected graph with vertices $\{y_0, \ldots, y_{d-1}\}$.
5: **for** $i = d$ to $n$ **do**
6:     $C \leftarrow$ GREEDY-SEARCH$(y_i, \{0\}, G, k, \ell^2\text{-distance})$.
7:     $N \leftarrow$ SELECT-NEIGHBORS$(0, C, d)$.
8:     Add edges $(y, z)$ to $G$ for every $z \in N$.
9:     **for** $z \in N$ **do**
10:         $C \leftarrow \{w \in \tilde{S} : (z, w) \text{ is an edge of } G\} \cup \{y\}$.
11:         $N \leftarrow$ SELECT-NEIGHBORS$(0, C, d)$.
12:         Let $N$ be the out-neighbors of $z$ in $G$.
13: $P' \leftarrow$ out-neighbors of 0 in graph $G$.
14: $P \leftarrow \{x_i \in S : y_i \in P'\}$.
15: Remove 0 and its incident edges from $G$ and replace the vertices of $G$ by the ones before transformation.
16: **Output:** $(G, P)$.

---

---

**Algorithm 3:** SELECT-NEIGHBORS$(x, C, d)$

---

1: **Input:** element $x$, the set of $k$-nearest neighbors $C$ of $x$, maximum outdegree $d$.
2: Initialize the out-neighbors set $N$ of $x$, i.e., $N \leftarrow \emptyset$.
3: Order $y_i \in C$ in ascending order of $\|x - y_i\|$.
4: $i \leftarrow 1$.
5: **while** $|N| \leq d$ and $i \leq |C|$ **do**
6:     **if** $\|x - y_i\| \leq \min_{z \in N} \|z - y_i\|$ **then**
7:         $N \leftarrow N \cup \{y_i\}$.
8:     $i \leftarrow i + 1$.
9: **Output:** a set of elements $N$.

---

isomorphic to a subgraph of $\ell^2$-Delaunay graph on $\tilde{S}$ after transformation. We will consider HNSW as an $\ell^2$-Delaunay graph approximation as proposed in [24]. The authors suggest that the hierarchy of Delaunay graph can be approximated by edge discrimination. Furthermore, we will consider a directed graph as an approximation to reduce the total degree. Given a dataset $\tilde{S}$, one wants to build the directed graph on $\tilde{S}$ iteratively. A directed graph is initialized by a random graph. In every iteration, for a given directed graph $G$ with vertices $\tilde{S}$, we consider an isolated vertex $x$ and apply greedy search (Algorithm 1) to find $k$-nearest neighbor of $x$, say $C_x$. $x$ will be connected to its nearest element, say $y_1$ in the candidate set $C_x$. Now the neighbor set is initialized to be $N(x) = \{y_1\}$. For the next nearest neighbor $y$, we add it to the neighbor set $N(x)$ if it satisfies edge selection

---
**Algorithm 4:** MIPS$(Q, S, K, k, l, d)$

---
1: **Input:** A set of queries $Q$, dataset $S$, the number of elements will be returned $K$, the size of candidate set $k$ for graph construction and $l$ for greedy search, maximum outgoing degree of graph $d$.
2: $(G, P) \leftarrow$ GRAPH-CONSTRUCTION$(S, k, d)$.
3: **for** $q \in Q$ **do**
4:     $C_q \leftarrow$ GREEDY-SEARCH$(q, P, S, G, l,$ inner product$)$.
5: **Output:** the set of top-$K$ objects $C_q \subset S$ in descending order of inner product with $q$ for $q \in Q$.

---

criterion: $\|x - y\| \leq \min_{z \in N(x)} \|z - y\|$. Iterative process stops when $d$ many valid neighbors are found. Algorithm 3 represents an embodiment of this procedure. This edge selection can improve the diversity of the direction of incident edges. We repeat this step and stop when either all elements in $C_x$ has been checked or the maximum outdegree $d$ is achieved. The edges $(x, y)$ for $y \in N(x)$ are added to the graph. Moreover, for $y \in N(x)$, we will add $x$ to $N(y)$. If $|N(y)| > d$, then we update $N(y)$ according to the edge selection criterion. This final step can reduce the effect caused by the random order of vertices. Corollary 1 suggests that IP-Delaunay graph is the neighborhood (in the graph sense) of 0 in $\ell^2$-Delaunay graph. So for any query $q$, we will apply greedy search starting from the out-neighbors of 0 (i.e., $P$ in Algorithm 2). Then the algorithm will search the optimal object w.r.t. inner product by greedy search. See Algorithm 4.

## 5 Experiments

In this section, we compare our method with state-of-the-art MIPS methods, on four common datasets (see Table 1): **Netflix**, Amazon Movie (**Amovie**) (http://jmcauley.ucsd.edu/data/amazon), **Yelp** (https://www.yelp.com/dataset/challenge) and **Music-100**. The first three are popular recommendation datasets. For Netflix, we use its 50-dimensional user and item vectors from [37]. For Amovie and Yelp, we utilize the matrix factorization method in [17] to get 100-dimensional latent vectors for user and item. Music-100 is introduced in [25] for the MIPS problem.

Table 1: Statistics of the datasets.

| Datasets | # Base Data | # Query Data | # Dimension | # Extreme | % Extreme |
|---|---|---|---|---|---|
| Netflix | 17770 | 1000 | 50 | 8017 | 45.12% |
| Amovie | 104708 | 7748 | 100 | 3169 | 3.03% |
| Yelp | 25815 | 25677 | 100 | 722 | 2.80% |
| Music-100 | 1000000 | 1000 | 100 | 304431 | 30.44% |

The ground truth of each query vector is the top-1, top-10, and top-100 measuring by the inner product. Only a fraction of data points can be the top-1 solution of (1), i.e., extreme points in Remark 1, whose percentage is an important feature of the dataset in MIPS problem. We estimate the percentage of extreme points for each dataset as below: for each vector $x$ in the base, we calculate its inner product $x^\top y$ with all vector $y$ in the base (including $x$ itself). Then we count the number of unique top-1 vector $y$ (i.e., extreme points) and compute the percentage of extreme points (i.e., last column of Table 1). This is not an exact estimation, but it is a tight lower bound.

### 5.1 Experimental Settings

We refer the new proposed algorithm as **Möbius-Graph**, and compare it with three previous state-of-the-art MIPS methods, **Greedy-MIPS** [37], **ip-NSW** [25], and **Range-LSH** [36], which are the most representative for MIPS. In Range-LSH, the dataset is first partitioned into small subsets according to the $\ell_2$-norm rank and then normalize data using a local maximum $\ell_2$-norm in each sub-dataset. This overcomes the limited performance due to the long tail distribution of data norms [36]. The authors of [37] used an upper bound of the inner product as the approximation of MIPS and designed a greedy search algorithm to find this approximation, called Greedy-MIPS. We use their original implementations. The open source code of ip-NSW adopts HNSW instead of NSW for graph construction. We found that the hierarchical structure and heuristic edge selection in HNSW does not significantly improve the performance of ip-NSW; see Figure 1. To provide comprehensive evaluation, we implement Möbius-Graph by both HNSW and SONG [39]. All comparing methods have tunable parameters. To get a fair comparison, we vary all parameters over a fine grid.

As the evaluation measures, we choose the trade-offs Recall vs. Queries Per Second (QPS) and Recall vs. Percentage of Computations. Recall vs. Queries Per Second reports the number of queries an algorithm can process per second at each recall level. Ideally, one wishes to have high recall levels, the method can process as many queries as possible (i.e., more efficient). Recall vs. Percentage of Computations checks the pair-wise computations at each recall level, the less the better. For each algorithm, we will have multiple points scattered on the plane by tuning parameters. To plot curves, we first find out the best result, $max_x$, along with the $x$-axis (i.e., Recall). Then 100 buckets are produced by splitting the range from 0 to $max_x$ evenly. For each bucket, the best result along the $y$-axis (i.e., the biggest amount of queries per second) is chosen. If there are no data points in the bucket, it will be ignored. In this way, we shall have at most 100 pairs of data for drawing curves. All experiments were performed on a 2X 3.00 GHz 8-core i7-5960X CPU server with 32GB memory.

## 5.2  Experimental Results

Experimental results for Recall vs. Queries Per Second (QPS) are shown in Figure 3. Each column corresponds to one dataset and figures in each row are results for top-1, top-10 and top-100 labels respectively. As can be seen, the proposed method Möbius-Graph works much better than previous state-of-the-art methods in most of the cases on all datasets.

The interesting fact is the effect of the extreme points percentage across different datasets. The Möbius-Graph embodiment is motivated by that the percentage of extreme points is low. As a result, the constructed approximate Delaunay graph would be efficient for maximum inner product retrieval. Nevertheless, we can see that, the proposed method works very well for datasets with a high percentage of extreme points, such as Netflix which has 45% of extreme points and the Music-100 which has more than 30%. We also show results for different ground truth label sets, which tell that the proposed method works well in various cases, not only for the top-1 label but also for the top-10 and top-100 labels. These results demonstrate the robustness of the proposed Möbius-Graph in MIPS.

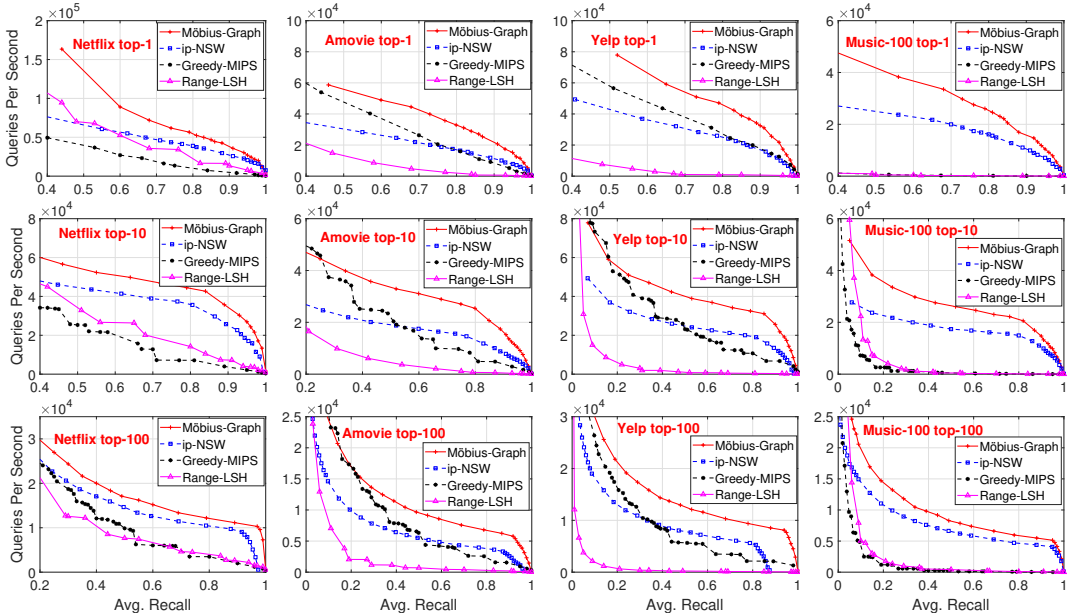

Figure 3: Experimental results for Recall vs. Queries Per Second on different datasets. We focus on top-1, top-10, and top-100 ground-truth labels. Here the best results are in the upper right corners.

Conversely, it is difficult to tell which baseline works better than others across all datasets. Range-LSH works relatively well on Netflix but much worse than other methods on the other three datasets. The baseline ip-NSW works well on datasets with high extreme points percentages (e.g., Netflix and Music-100) but becomes worse on other datasets. Greedy-MIPS shows priorities over ip-NSW on datasets with low extreme points percentages (e.g., Amovie and Yelp) at some recall levels.

Results for Recall vs. Percentage of Computations are shown in Figure 4. Only top-10 results are shown due to the limited space. Top-1 and top-100 results can be found in the Appendix. Note that this measurement is not meaningful for Greedy-MIPS. Results for Recall vs. Percentage of Computations are shown in Figure 4. In this view, the proposed Möbius-Graph works best in all

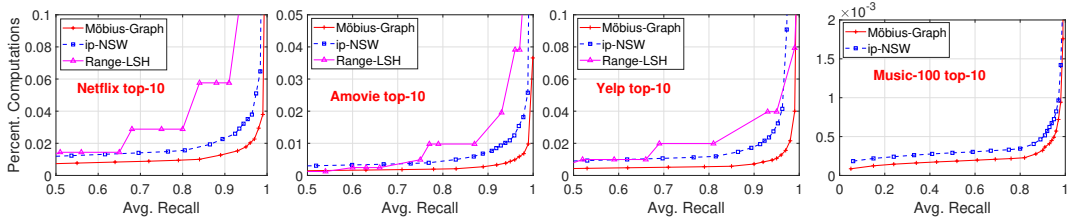

Figure 4: Experimental results for Recall vs. Percentage of Computations on different datasets. Best results are in the lower right corners.

cases. Range-LSH works comparably with others on smaller datasets (i.e., the first three) in this view. Recall vs. Percentage of Computations does not consider the cost of different index structures. Although Range-LSH works well in this view, its overall time cost is much higher than others as shown in Figure 3. The possible reason is that the table based index used in Range-LSH is not that efficient in searching. Besides, Range-LSH works badly on Music-100, which is much larger. The curve for Range-LSH cannot be shown in the scope of Music-100.

Besides, we represent the graph construction time cost by ip-NSW and Möbius-Graph in Table 2. As can be seen, Möbius-Graph consumes 13.7% to 65.5% less time in index construction than ip-NSW, which brings great benefits for real applications. The reason is that metric measure (i.e., $\ell_2$) based searching (in the graph construction) is more efficient than inner product based searching.

Table 2: Graph Construction Time in Seconds.

|  | Netflix | Amovie | Yelp | Music-100 |
|---|---|---|---|---|
| ip-NSW | 2.19 | 36.95 | 6.78 | 396.82 |
| Möbius-Graph | 1.89(-13.7%) | 24.35(-34.1%) | 2.34(-65.5%) | 162.24(-59.1%) |

## 5.3 Implementation by SONG

To exclude bias from implementation, we also implement Möbius-Graph and ip-NSW by another search on graph platform, SONG [39]. The results are shown in Figure 5. As can be seen, the implementation of SONG is more efficient than HNSW, both for Möbius-Graph and ip-NSW, but their priority order keeps the same. Möbius-Graph works better than ip-NSW under both implementations.

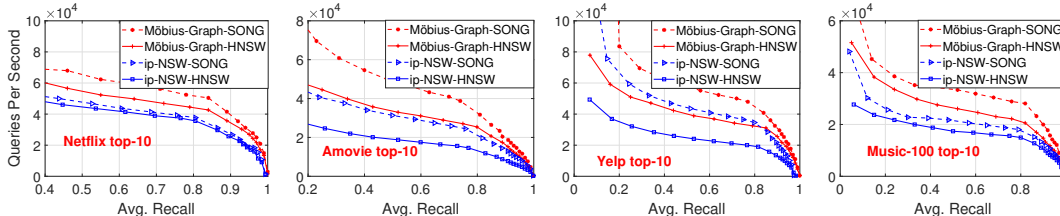

Figure 5: Comparison of two implementations, HNSW and SONG, on Möbius-Graph and ip-NSW.

## 6 Conclusion and Future Work

Maximum Inner Product Search (MIPS) is a challenging problem with wide applications in search and machine learning. In this work, we develop a novel search on the graph method for MIPS. In the view of computational geometry, we show that under Möbius transformation, an isomorphism exists between Delaunay graph for inner product and $\ell^2$-norm. Based on this observation, we present a graph indexing algorithm that converts subgraph of $\ell^2$-Delaunay graph into IP-Delaunay graph. Then, we perform MIPS via greedy search on the transformed graph. We demonstrate that our approach provides an effective and efficient solution for MIPS.

This paper focuses on fast search under the non-metric measure, inner product. Beyond inner product, more complicated measures has been studied, such as Bregman divergence [6], max-kernel [8, 7] and even more generic measures [32]. It would be interesting to extend the method proposed in this paper to these measures. Another promising direction is to adopt a GPU-based system for fast ANN search and MIPS, which has been shown highly effective for generic ANN tasks [22, 19, 39]. Developing GPU-based algorithms for MIPS (and related applications) is a topic which can be further explored.

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
