[Supplementary Material]

# Supplementary Material to " Möbius Transformation for Fast Inner Product Search on Graph"

**Zhixin Zhou, Shulong Tan, Zhaozhuo Xu and Ping Li**
Cognitive Computing Lab
Baidu Research
10900 NE 8th St. Bellevue, WA 98004, USA
1195 Bordeaux Dr. Sunnyvale, CA 94089, USA
{zhixin0825,zhaozhuoxu}@gmail.com, {shulongtan,liping11}@baidu.com

## A  Search on Delaunay Graph

Greedy search on Delaunay graph is sufficient and necessary for achieving the global optimum in ANN search [1, 2]. The sufficiency is generalized to a larger class of $f$ in [4]. We first consider a general optimization problem. Let $X \subset \mathbb{R}^d$, we consider a data set $S = \{x_1, \ldots, x_n\} \subset X$ and aim to solve the optimization problem, for $q \in X$,

$$\arg\max_{x_i \in S} f(x_i, q) \quad \text{where} \quad f : X \times X \to \mathbb{R}. \tag{1}$$

Assuming $f$ is continuous, we have the following theorem.

**Theorem 2.** *For given $f$, we assume for any dataset $S$, each the Voronoi cell $R_i$ is a connected. Let $G = (S, E)$ be the Delaunay graph w.r.t. the Voronoi cells. Then for any $q \in X$, simple greedy search on Delaunay graph returns the solution of* (1). *In other words, let $N(x_i) = \{x_j \in S : \{x_i, x_j\} \in G\}$ be the neighbors of $x_i$ on Delaunay graph. If $x_i$ satisfies*

$$f(x_i, q) \geq \max_{x_j \in N(x_i)} f(x_j, q), \tag{2}$$

*then $x_i$ a solution of* (1). *Conversely, for any $G'$ does not contain Delaunay graph as a subgraph, there exists a query $q \in Y$ such that greedy search on $G'$ does not always retrieve all global maximum.*

*Proof.* By the assumption on $f$, we have

$$\tilde{R}_i = \bigcap_{x \in N(x_i)} \{q \in X : f(x_i, q) \geq f(x_j, q)\}$$

is connected and $R_i \cup \{q\} \subset \tilde{R}_i$. Hence we can define a path $c : [0, 1] \to \mathbb{R}^k$ such that $c(0) \in R_i$ and $c(1) = q$. For every $x_j \in S$, $f(x_j, c(0)) \leq f(x_i, c(0))$. If $f(x_j, c(1)) \geq f(x_i, c(1))$, then by intermediate value theorem, there exists $t \in [0, 1]$ such that $f(x_j, c(t)) = f(x_i, c(t))$. Hence $R_i \cap R_j \neq \emptyset$, and $x_j$ is a neighbor of $x_i$ on $G$. In this case, by (2), we must have $f(x_i, q) = f(x_j, q)$. Therefore, for $x_j \in S$, we have $f(x_i, q) \geq f(x_j, q)$.

Conversely, suppose $G'$ misses an edge in Delaunay graph, then there exists $x_i$ and $x_j$ such that $R_i \cap R_j \neq \emptyset$, but $x_j \notin N(x_i)$. Suppose the query $q \in R_i \cap R_j$ and the initial point is $x_i$, then both $x_i$ and $x_j$ are global maximum of $f(\,\cdot\,, q)$. $x_j$ is not neighbor $x_i$, but $x_i$ is a global maximum, so greedy search stops at this step. Thus, $x_j$ cannot be discovered as a global maximums. □

## B  Additional Comments on Assumption 1

Assumption 1 eases the arguments in Section 3. For better understanding of Assumption 1, we develop the following equivalent expressions.

**Proposition 3.** *The following are equivalent:*

*(a) $S$ satisfies (A1).*

*(b) The convex hull of $S$ contains 0 as an interior point.*

*(c) For every $a \in \mathbb{R}^d \backslash \{0\}$, there exists $x \in S$ such that $x^\top a > 0$.*

*Proof.* (a) $\Rightarrow$ (b). Suppose 0 is not an interior point of $\mathrm{Conv}(S)$, then there exists a closed half-space $H$ with a boundary point 0 contains $S$. $H$ is a convex cone, so $\mathrm{coni}(S) \subset H \subsetneq \mathbb{R}^d$.

(b) $\Rightarrow$ (c). For every $a \in \mathbb{R}^d \backslash \{0\}$, there exists $\beta > 0$ such that $\beta a \in \mathrm{Conv}(S)$. Hence $\beta a = \sum_{i=1}^n \alpha_i x_i$ for some $\alpha_i \geq 0$. Then $0 < \beta a^\top a = \beta a^\top \sum_{i=1}^n \alpha_i x_i = \beta \sum_{i=1}^n \alpha_i x_i^\top a$, so there exists $x \in S$ such that $x^\top a > 0$.

(c) $\Rightarrow$ (a). Suppose $\mathrm{coni}(S) \neq \mathbb{R}^d$, then $\mathrm{coni}(S) \subset H$ for some closed half-space $H$. For $a \notin H$ such it is perpendicular to the boundary $H$, there does not exists $x \in S$ such that $x^\top a > 0$. $\quad\square$

Suppose Assumption 1 is not satisfied, the MIPS problem is still interesting. We will discuss this situation in the following two cases.
**Case 1.** If Assumption 1 is not true, but the queries always locate in the conical hull of the dataset $S$, then our approach is still valid because, for every query, the correct solution of MIPS problem is still a neighbor of 0 after Möbius transformation.

**Case 2.** Suppose Assumption 1 is not true, and queries can be any points in the Euclidean space, then our approach does not work. However, we can slightly change the graph construction algorithm as follows. We find the center of the dataset, say $c$, then we apply the transformation

$$g(x) = \frac{x - c}{\|x\|^2}$$

to every data point to obtain $\tilde{S}$. We note that such $g(x)$ is still a Möbius transformation since it is of the form in (3). It is not difficult to check the isomorphism between IP-Delaunay graph and the subgraph of $\ell^2$-Delaunay graph introduced in Corollary 1. However, this method is only suggested in this special case. Centering the data points changes all the norms, while the length of the vector decides the chance of being returned in MIPS problem.

## C   Proof of Theorem 1

(a) $\Rightarrow$ (b). By Definition 1, the Voronoi cell $R_i$ w.r.t. inner product and $x_i$ is

$$R_i = \{q \neq 0 : x_i^\top q \geq x_k^\top q \text{ for } k \in [n]\}.$$

Similarly,

$$R_j = \{q \neq 0 : x_j^\top q \geq x_k^\top q \text{ for } k \in [n]\}.$$

By Definition 2, (a) implies there exists $a \in R_i \cap R_j$. $a$ also satisfies $x_i^\top a = x_j^\top a \geq \max_{x \in S} x^\top a > 0$.

(b) $\Rightarrow$ (a). If $a$ satisfies statement (b), then $a \in R_i \cap R_j$, which implies (a) by Definition 2.

(b) $\Rightarrow$ (c). Firstly, we notice that $x_i = y_i / \|y_i\|^2$, then we let $b = x_i^\top a$ and $c = \frac{a}{2b}$. We note that $b > 0$ by Proposition 3 (c). Then we have

$$y_i^\top c = \frac{y_i^\top a}{2x_i^\top a} = \frac{y_i^\top a \|y_i\|^2}{2y_i^\top a} = \frac{1}{2}\|y_i\|^2.$$

Hence, $\|y_i - c\|^2 = \|y_i\|^2 - 2y_i^\top c + \|c\|^2 = \|y_i\|^2 - \|y_i\|^2 + \|c\|^2 = \|c\|^2$. Using (A1), we have $x_i^\top a \geq \max_{x \in S} x^\top a > 0$, so for $x \in S$ and $y = x/\|x\|^2 \in S'$,

$$y^\top c = \frac{y^\top a}{2x_i^\top a} = \frac{x_j^\top a \|y_j\|^2}{2x^\top a} \leq \frac{1}{2}\|y_j\|^2.$$

Therefore, $\|y - c\|^2 = \|y\|^2 - 2y^\top c + \|c\|^2 \le \|y\|^2 - \|y\|^2 + \|c\|^2 = \|c\|^2$. Since this is true for all $x \in S$, we have $\|c\| \le \min_{y \in S'} \|y - c\|$. Since $x_i^\top a = x_j^\top a$, we can repeat the arguments for $x_j$ to obtain statement (c).

(c) $\Rightarrow$ (b). This can be proved by observing that every step of the proof of (b) $\Rightarrow$ (c) is invertible.

(c) $\Leftrightarrow$ (d). This is due to empty sphere criterion. See Proposition 2.

## D   Additional Empirical Experiments

In [4], it was claimed that their algorithm can adopt any graph construction algorithm, including NSW [2] and HNSW [3]. For the sake of fairness, we compare **Möbius-Graph** with different versions of **ip-NSW**. Edge selection is a novel contribution of HNSW. However, there is no guarantee for its applicability on non-metric measure. We compare the MIPS efficiency of ip-NSW with and without edge selection step and find an interesting observation. Figure 6 shows that, for Amovie and Yelp, edge selection results in poor performance, while the effect is not obvious on the other two datasets. It is possible that edge selection is not helpful when the proportion of extreme points is small.

The hierarchical graph structure in HNSW is to perform multi-scale hopping. Our Möbius-Graph can find good starting points, so it would be interesting to see whether ip-NSW can work well by starting points found by Möbius-Graph. Here we design one variant for ip-NSW, **ip-NSW-init**, which gives up the hierarchical index structure but exploits starting points found by Möbius-Graph. For each query, we will exploit Möbius-Graph to find a start point by conducting one step greedy search. This step is done offline and the time cost will not be counted as that of ip-NSW-init. The results are represented in Figure 7. If ip-NSW starts searching from initial points found by Möbius-Graph, its performance can be significantly improved in top-1 inner product search. However, such difference disappears if we consider top-100 results. We also compare the effect of hierarchical graph structure on the performance of ip-NSW. As can be seen in Figure 8, its impact is very little.

Figure 6: Experimental results for Möbius-Graph, ip-NSW with and without edge selection.

Figure 7: Experimental results for Möbius graph, ip-NSW using random initial points and ip-NSW using initial points from Möbius graph.

Figure 8: Experimental results for Möbius-Graph, ip-NSW with and without hierarchical structure.

Figure 9 completes the experimental results of Figure 4 in Section 5.

Figure 9: Experimental results for Recall vs. Percentage of Computations. We show remaining results for top-1 and top-100 labels. The curves for Range-LSH on Music-100 are out of the showing scopes.