[Reviews · NeurIPS 2019]

Reviewer 1



This paper proposed a new algorithm for max-inner-product-search, a widely encountered problem in all kinds of applications. Though seemingly similar to ANN problem, MIPS is different in terms of theory and algorithm design, so that the massive amount of KNN methods cannot apply directly. The authors extend the well-known Delaunay graph type of methods in ANN to MIPS and provide both theoretical discussion and experimental evidence to show the advantage of the proposed method. I find this paper to be interesting, and would like the authors to consider my following comments: 1. For assumption 1, I'm a little confused. It is not clear to me why this assumption is important. In general, the data points can be shifted without causing problems, so that conical hull should be the whole space as long as the data points are in general position. 2. Line 126 "dataset approximately follow the normal distribution" is not generally true. The datasets can easily be multi-Gaussian or more complicated, but of course this won't affect the paper's assumption that 0 is an interior point of the dataset. Maybe a better way of expression is to generally check this assumption in the datasets used in the experiments. Also, it would be nice to see how the algorithm is affected if assumption 1 does not hold. 3. A trend of the ANN and MIPS community is to involve GPU for acceleration. Though it's a matter of hardware and implementation, nowadays the algorithm design is deeply influenced so that an algorithm with a little higher complexity but more compatible with GPU is preferred. In some cases, GPU+exact search is already as fast as approximate search. The scale of some of the datasets used in this paper already fall into this scenario. The authors may consider to involve some of the related literatures like [1-3] or discuss the possibility of making the algorithm easier for parallelization or compatible with GPU. 4. The experiments in section 5. I'm confused about the running time experiments. I only find QPS test but not preprocessing time. Since approximate type of search methods usually require pre-processing and the overhead is generally large, it is better to also demonstrate the latency comparison for that part. 5. The paper's contribution might be a concern since it is an extension from exiting algorithms. But at least the experiments show positive results so it might still be meaningful and it is indeed an interesting work. [1] Johnson, J., Douze, M. and Jégou, H., 2019. Billion-scale similarity search with GPUs. IEEE Transactions on Big Data. [2] Wang, C., Tang, L., Bian, S., Zhang, D., Zhang, Z. and Wu, Y., 2019, May. Reference Product Search. In Companion Proceedings of The 2019 World Wide Web Conference (pp. 404-410). ACM. [3] Li, S. and Amenta, N., 2015, October. Brute-force k-nearest neighbors search on the GPU. In International Conference on Similarity Search and Applications (pp. 259-270). Springer, Cham. ---------------- Update: Thanks to the authors for the response. I think the rebuttal has cleared my confusions and some issues mentioned in my previous comments. I decide to change my rating to 7. One further comment regarding assumption 1: you only need to perturb the dataset a little if you have a zero vector in the dataset (which is usually the case in practice). Or more directly, if you add (0, 0, ..., 0, +-\epsilon, 0, ..., 0) to the original dataset, it won't affect the MIPS task since the added vectors have very small norm, but then assumption 1 becomes true automatically. I'm just trying to understand what is actually needed here for the theories to hold.

Reviewer 2



This paper proposes a new approximated search method which improves a recently proposed graph-based maximum inner product search (MIPS) method. The idea is to use Mobius transformation to convert an inner product problem into a l^2 problem, and then use HNSW to solve it. This idea is simple yet ingenious. In this way, the advantageous components of HNSW, an approximation of l^2-Delaunay graph, can fully function in the IP setting. Experiments show that the proposed method significantly outperforms the state-of-art methods. The writing is clear and easy to follow. A good paper.

Reviewer 3



This paper studies the Maximum Inner Product Search (MIPS) problem. MIPS is vital for various machine learning tasks such as Recommender Systems and multi-label classification. This paper proposed a new graph-based index for fast MIPS. The main idea is to apply Mobius transformation on data and then construct a l2 distance proximity graph (on transformed data). Then MIPS is conducted by greedy search on this graph. The authors provide theoretical analysis for the proposed method. They prove that the construct proximity graph in l2 on Mobius transformed data is an isomorphism of the IP-Delaunay graph on original data, which is the theoretical foundation of conducting MIPS on graph-based index. Strong points: (1) The idea of processing data by Mobius transformation is impressive and make sense for the MIPS task. After transforming, MIPS answers (those with higher norms) would be near to 0. So, the additional data point, 0, provides good initial nodes for the greedy search. (2) Comparing to the previous graph-based MIPS method, ip-NSW, this paper provides more solid theoretical analysis for the proposed method, which is also a key part of this paper's contribution. (3) The proposed method is evaluated by extensive experiments on various public datasets. The baselines are all state-of-the-art methods for the MIPS problem, published in recent two years. The improvement is significant in most cases. My suggestions: (1) The disadvantages of ip-NSW explained in Section 2 are validated by experiments in Appendix G, this is good, but I would suggest to move part of related experimental results to Section 2 from appendix. This part is important for readers to understand the previous method ip-NSW and motivations of the proposed method. (2) Typos. I found in some figures of Appendix, the proposed method is referred as 'Mobius' but as 'Mobius-graph' in other figures. After the authors' response: All my questions are well answered. I would not change my score.

[Author Response · NeurIPS 2019]

# 1 Response to Reviewer 1

**Re assumption 1:** Shifting the data points is a good idea, but it might cause problems. Shifting the data points changes the norms of all vectors, while the norms are very important quantities in the MIPS problem. Without shifting, the greedy search algorithm exploits from well-chosen top-norm vectors, but this advantage is no longer valid after shifting.

In our current work, we focus on theory and datasets satisfying assumption 1. Our approach shows improvement in these datasets. We have not implemented our algorithm on datasets violating assumption 1, so we did not make a conclusion about shifting. Indeed, we have not found any datasets violating assumption 1 in recommendation system, but we will further explore related experiments in artificial datasets in our future work. Further discussions about assumption 1 and shifting appear in Appendix C in the supplemental file.

**Re normal assumption:** Thanks for pointing this out. Normal distribution assumption is much more than enough and might not be realistic. We will rephrase the sentence as follows: "In these scenarios, the entries of data vectors distributed on the whole real line. With high probability, each hyperoctant contains at least one data point so that the convex hull of the dataset contains 0 as an interior point."

**Re GPU acceleration:** Thanks for your suggestions on the relevant literature. We will definitely add them to our reference and further discuss their work in the section of related work. Indeed, GPU+exact search performs well on some datasets, but we believe approximate search methods have the advantage in scalability on larger datasets.

In the present work, we aim to improve the efficiency of the MIPS problem in algorithmic perspective. However, the implementation of our method on GPU-platform can be an interesting and novel extension. GPU can process multiple queries in parallel. GPU streaming multiprocessors can compute inner products very fast. Nevertheless, in our approach, the set of priority queue (set $C$ in Algorithm 1) and the indices of visited vertices can be arbitrarily large. Theoretically, their sizes only have a trivial upper bound, the number of total vertices $n$. It does not hurt the efficiency for implementation on CPU, but straightforwardly transplanting this algorithm to the GPU is problematic. We will add extra discussion in the paper and leave the details for future work.

**Re preprocessing time:** In our experiments, preprocessing time only includes graph construction time, which is presented in Table 2, at the end of Section 5.3, in line 281. We compare the graph construction time with ip-NSW. As can be seen in the table, our approach consumes much less time during the preprocessing procedure. The table is duplicated as follows:

Table 2    Graph Construction Time in Seconds.

|  | Netflix | Amovie | Yelp | Music-100 |
|---|---|---|---|---|
| ip-NSW | 2.19s | 36.95s | 6.78s | 396.82s |
| Möbius-Graph | 1.89s(-13.7%) | 24.35s(-34.1%) | 2.34s(-65.5%) | 162.24s(-59.1%) |

**Re contribution:** Thanks for finding out our work is interesting. We agree our work is based on existing approximate nearest neighbor (ANN) search on graph algorithms, which have been studied for about 15 years. It has been improving and applying to various search tasks. It was shown that well-designed heuristics can significantly improve searching performance for ANN search, but generalization of these ideas to the MIPS is non-trivial. The goal of our paper is to fill this gap.

# 2 Response to Reviewer 2

Thank you so much for highly encouraging comments. We will change the sizes of markers in Figure 2 and Figure 3. We address your concern about the normal assumption in our response to reviewer 1. The normal assumption is indeed not necessary. We will rephrase the sentence more accurately.

# 3 Response to Reviewer 3

We appreciate your detailed nice summary of our work. We will move some plots from Figure 5 and 6 to Section 2 if space allows. We will also change "Möbius" to "Möbius-graph" in Figure 5, 6 and 7.



[Meta-Review · NeurIPS 2019]

There is a consensus among reviewers that this paper is well above the acceptance threshold. The rebuttal did move the opinion in positive directions. Authors are encouraged to take reviews into account to prepare the final version. [This meta-review was reviewed and revised by the Program Chairs]